# Preliminary Estimates of Age, Growth and Natural Mortality of Margate, *Haemulon album*, and Black Margate, *Anisotremus surinamensis*, from the Southeastern United States

**Michael L. Burton \*, Jennifer C. Potts and Andrew D. Ostrowski**

National Oceanic and Atmospheric Administration, National Marine Fisheries Service, Southeast Fisheries Science Center, Beaufort Laboratory, 101 Pivers Island Road, Beaufort, NC 28516, USA

\* Correspondence: Michael.Burton@noaa.gov; Tel.: +1-252-728-8756

**Abstract:** Ages of margate, *Haemulon album* ($n$ = 415) and black margate, *Anisotremus surinamensis* ($n$ = 130) were determined using sectioned sagittal otoliths collected from the Southeastern United States Atlantic coast from 1979 to 2017. Opaque zones were annular, forming between January and June for both species, with peaks in occurrence of otoliths with opaque margins in April for margate and March for black margate. The observed ages for margate were 0–22 years, and the largest fish measured 807 mm TL (total length). Black margate ranged in age from 3 to 17 years, and the largest fish was 641 mm TL. Weight–length relationships were: margate, $\ln(W) = 2.88 \ln(TL) - 10.44$ ($n$ = 1327, $r^2$ = 0.97, MSE = 0.02), where $W$ is total weight (grams, g); black margate, $\ln(W) = 3.02 \ln(TL) - 11.10$ ($n$ = 451, $r^2$ = 0.95, MSE = 0.01). Von Bertalanffy growth equations were $L_t = 731 (1 - e^{-0.23(t+0.38)})$ for margate, and $L_t = 544 (1 - e^{-0.13(t+2.61)})$ for black margate. After re-estimating black margate growth using a bias-correction procedure to account for the lack of younger fish, growth was described by the equation $L_t = 523 (1 - e^{-0.18(t+0.0001)})$. Age-invariant estimates of natural mortality were M = 0.19 $y^{-1}$ and $M$ = 0.23 $y^{-1}$ for margate and black margate, respectively, while age-varying estimates of $M$ ranged from 2.93 −0.23 $y^{-1}$ for fish aged 0–22 for margate and 7.20 − 0.19 $y^{-1}$ for fish aged 0–18 for black margate. This study presents the first documentation of life-history parameters for margate from the Atlantic waters off the Southeastern United States, and the first published estimate of black margate life history parameters from any geographic region.

**Keywords:** age; growth; natural mortality; margate; *Haemulon album*; black margate; *Anisotremus surinamensis*

## 1. Introduction

Margate (*Haemulon album* Cuvier 1830) and black margate (*Anisotremus surinamensis* Bloch 1790) are large-sized members of the grunt family (Family Haemulidae), capable of attaining lengths of 750 mm and 600 mm, respectively. They have similar geographic ranges in the tropical Western Atlantic, with margate distributed from Southeastern Florida through Brazil, including throughout the Gulf of Mexico to the Yucatan and throughout the Antilles [1]. Black margate are distributed from Cape Canaveral, Florida, south to the Bahamas, throughout the Gulf of Mexico to Cuba and throughout the Caribbean, southward to Brazil [1]. The two species have similar habitat associations, with late-stage larvae and juveniles utilizing shallow tidal flats, seagrass beds, mangroves, and nearshore hardbottom habitats [1–3]. As both species grow from juveniles to adults, they move to transitional hardbottom and reef habitat [4], moving to deeper offshore reefs as they grow larger [5]. Both species have been found at depths of 60 m [6,7].

In U.S. territorial waters, margate and black margate are species of moderate economic importance to recreational anglers primarily in the Florida reef fish fishery. Estimated annual landings of margate caught by private recreational and charterboat anglers from Florida averaged 71,253 fish at 25,554 kg for 1981–2017, while black margate landings averaged 117,664 fish at 75,553 kg for the same time period [8]. Landings from other states in the Southeastern U.S. (SEUS, North Carolina through the Dry Tortugas Florida) were negligible. Anglers fishing from headboats (vessels carrying seven or more anglers engaged in recreational fishing) sampled by the Southeast Region Headboat Survey (SRHS) in Florida, averaged 1267 margate (2347 kg) and 283 black margate (329 kg) annually, during 1981–2017 [9]. Except for two years where private recreational-charterboat landings of black margate were unusually high (2000, 2009), combined recreational landings exhibit no consistent trends for either species. Commercial landings of margate from Florida waters averaged 1174 kg annually from 1995 to 2013, while commercial fishermen from North Carolina averaged 4900 kg of margate landings from 1994 to 1998 [10]. Reported commercial landings of black margate in the SEUS for 1981 to 2017 were negligible.

Margate are currently included in the South Atlantic Fishery Management Council's (SAFMC) Snapper-Grouper Fishery Management Plan (FMP) [11]. Management is by inclusion in an aggregate all-species bag limit of 20 fish per person per day in the recreational fishery. There are no recreational size limits or commercial size or trip limits. Black margate were previously included in the Snapper-Grouper FMP but were removed in 2012. The only current management restriction on black margate in the SEUS is a 100 pound possession bag limit in Florida state waters [12].

We undertook the study of these two species because of the need for basic biological information on all data-limited species. The only previous studies of the biology of either of these species were an age-growth study of margate from Jamaica that used scales as the ageing structure [13] and a study from Cuba using otoliths [14]. While these two species are unlikely to become the subject of a stock assessment by the Southeast Data Assessment and Review (SEDAR), the process by which federally managed species are prioritized and assessed for stock assessments in the SEUS, basic life history information for use in other management efforts (e.g., ecosystem-based fishery management) is still needed. Our analyses relied on archived sagittal otoliths collected by long-term, systematic dockside sampling programs. Our primary goal is to provide updated and comprehensive information on age-growth parameters and natural mortality rates for margate and black margate from the SEUS, filling an important gap for these data-limited species.

## 2. Results

### *2.1. Age Determination and Timing of Opaque Zone Formation*

A total of 415 margate sagittae and 130 black margate sagittae were available from archived samples and were sectioned. Opaque zones were counted on 100% of sections for both species. The majority of samples of both species came from Florida. One margate was collected from North Carolina and two from East Central Florida, with the remaining fish coming from Southeast Florida through the Dry Tortugas ($n = 412$, or 99%). All black margate samples came from Florida, with 94% ($n = 122$) coming from southeast Florida or the Florida Keys, and eight fish collected from east central Florida. The majority of samples for both species came from recreational fisheries (margate: 83%; black margate: 68%). Commercially reported catches for both species came primarily from hook-and-line fisheries and secondarily, from spear fishermen.

Opaque zones on sectioned otoliths of both species were legible (Figure 1), and the between-reader index of average percent error (IAPE) was 5.42% ($n = 194$, or 47% of sections) for margate and 3.57% ($n = 27$, or 20% of sections) for black margate, values close to or meeting the acceptable criteria of 5% for species of moderate longevity and reading complexity for IAPE values [15]. Direct agreement between readings was 58% and 48% for margate and black margate, respectively. Agreement to ±1

year was 94% and 85%, respectively. We examined the entire section when assigning counts, but the most consistent ones were obtained from the dorsal portion and along the sulcal groove.

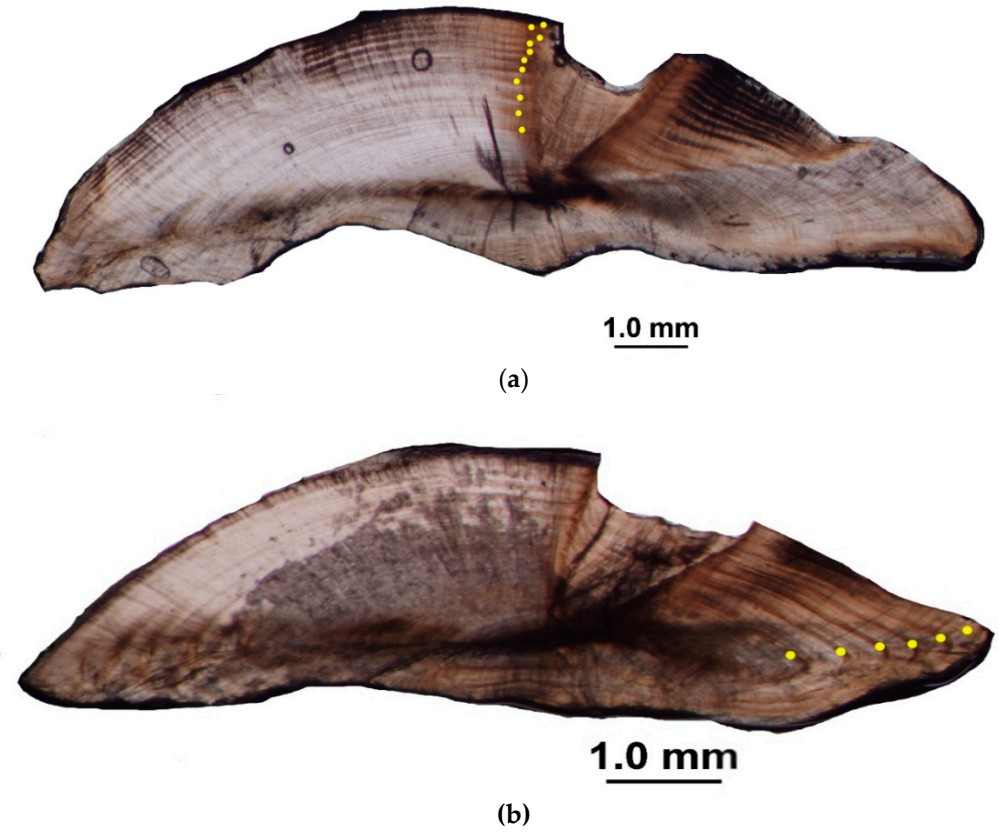

**Figure 1.** Sections from otoliths of (**a**) margate (*Haemulon album*), 586 mm total length (TL), 11-yr; (**b**) black margate (*Anisotremus surinamensis*), 357 mm TL, 6-yr.

We were able to assign an edge type to all aged samples for both species for our analysis of opaque zone formation timing. Margate otoliths exhibited opaque zones on the edge January through June, with a peak in the occurrence of such otoliths in February (Figure 2). A shift to a narrow translucent edge was prominent in July and August, followed by an increase in the occurrence of sections with moderate translucent edges in September, and the occurrence of wide translucent edge types increasing from October to January, when opaque edges begin to form. Margate otoliths exhibited no opaque zones on the edge from July through December. Black margate otoliths exhibited very similar edge patterns. Opaque zones were found on the edges of otolith sections from January through June, with a peak occurrence in March (Figure 3). Narrow translucent edges were predominant from June through August, and moderate to wide translucent edges become more common from September through December. The occurrences of sections with wide translucent edge types decreased from January through March in concert with the increase in opaque zones on the edge. We concluded that opaque zones on both margate and black margate otoliths were annuli.

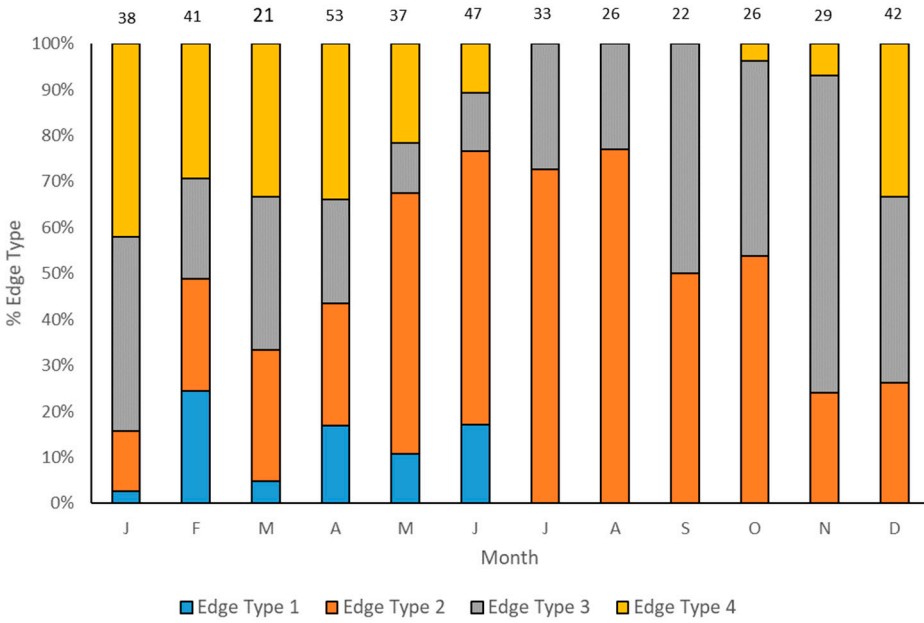

**Figure 2.** Monthly percentages of all edge types for margate (*Haemulon album*) collected from the Southeastern United States from 1979 to 2017, with total sample sizes above each column. Edge codes: 1 = opaque zone on edge, indicating annulus formation; 2 = small translucent zone, <30% of previous increment; 3 = moderate translucent zone, 30%–60% of previous increment; 4 = wide translucent zone, >60% of previous increment [16].

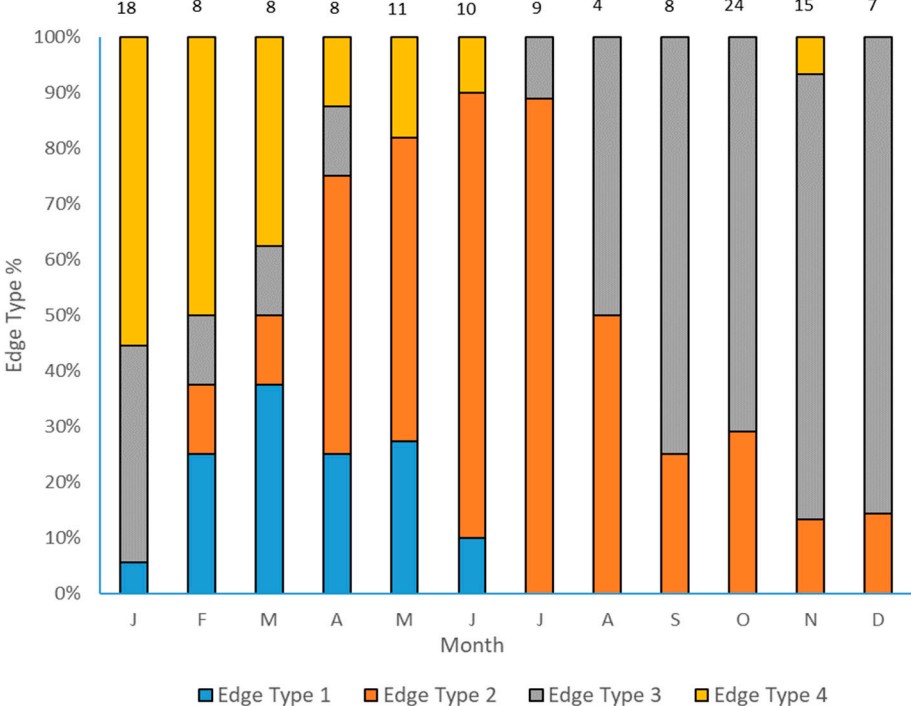

**Figure 3.** Monthly percentages of all edge types for black margate (*Anisotremus surinamensis*) collected from the Southeastern United States from 1979 to 2017, with total sample sizes above each column. Edge codes: 1 = opaque zone on edge, indicating annulus formation; 2 = small translucent zone, <30% of previous increment; 3 = moderate translucent zone, 30%–60% of previous increment; 4 = wide translucent zone, >60% of previous increment [16].

Based on the above-reported data on timing of opaque zone formation, we assigned fish to their year class, or calendar ages, for both species in the following manner: for fish caught January through June and having an edge type of 3 or 4, the annuli count was increased by one; for fish caught in that same time period with an edge type of 1 or 2, calendar age was equivalent to annuli count; similarly, for fish caught from July to December, the calendar age was equivalent to the annuli count.

## 2.2. Growth

Margate ranged in size from 221 to 807 mm TL and in age from 0 to 22 years (Table 1), but only four fish were older than 17 years of age. Parameter estimates (standard error, 95% confidence intervals) of the von Bertalanffy growth model [17], estimated using all samples, were $L_\infty$ = 731 (15.8, 700–762), $k$ = 0.23 (0.02, 0.20–0.27), and $t_0$ = −0.38 (0.18, −0.74−−0.03); $n$ = 415, Figure 4.

Observed ages fit the predicted growth curve well. Margate grew fairly rapidly, achieving a mean observed size of 582 mm TL (80% of $L_\infty$) by age-6. Growth slowed through age-12, achieving a mean observed size of 708 mm TL (97% of $L_\infty$). Margate then grew an average of 4 mm per year from age 13 through age-22.

Black margate ranged from 241 to 641 mm TL and ages 3–18 years (Table 2). Initial parameter estimates of the von Bertalanffy growth model were $L_\infty$ = 544 (77, 392–696), $k$ = 0.13 (0.07, 0.01–0.27), and $t_0$ = −2.61 (2.99, −8.54−3.31); $n$ = 130, Figure 5.

**Table 1.** Observed and predicted mean total length (TL) from the von Bertalanffy growth model, measured in millimeters, and natural mortality at calendar-age [18] data for margate (*Haemulon album*) collected from 1979 to 2018 from Southeast Florida. Standard errors of the means (SE) are provided in parentheses.

| Age | $n$ | Mean Observed TL (±SE) | TL Range | Predicted TL | $M$-$y^{-1}$ |
|-----|-----|-----|-----|-----|-----|
| 0 | 1 | 222 | – | 64 | 2.93 |
| 1 | 3 | 258 (4) | 250–265 | 202 | 1.11 |
| 2 | 42 | 313 (7) | 246–411 | 311 | 0.68 |
| 3 | 81 | 384 (7) | 260–512 | 398 | 0.51 |
| 4 | 97 | 465 (6) | 290–595 | 467 | 0.42 |
| 5 | 66 | 520 (10) | 285–672 | 522 | 0.36 |
| 6 | 49 | 582 (9) | 415–722 | 565 | 0.32 |
| 7 | 21 | 615 (11) | 460–684 | 599 | 0.30 |
| 8 | 12 | 647 (18) | 532–710 | 626 | 0.28 |
| 9 | 4 | 666 (34) | 595–730 | 648 | 0.27 |
| 10 | 8 | 639 (33) | 476–735 | 665 | 0.26 |
| 11 | 6 | 683 (23) | 586–750 | 679 | 0.25 |
| 12 | 4 | 708 (32) | 647–790 | 690 | 0.25 |
| 13 | 5 | 712 (20) | 666–757 | 698 | 0.24 |
| 14 | 4 | 676 (11) | 647–700 | 705 | 0.24 |
| 15 | 4 | 687 (49) | 570–807 | 710 | 0.24 |
| 16 | 1 | 616 | – | 715 | 0.24 |
| 17 | 3 | 689 (20) | 651–719 | 718 | 0.24 |
| 18 | 1 | 699 | – | 721 | 0.23 |
| 19 | | – | – | 723 | 0.23 |
| 20 | 2 | 694 (9) | 685–703 | 725 | 0.23 |
| 21 | | – | – | 726 | 0.23 |
| 22 | 1 | 710 | – | 727 | 0.23 |

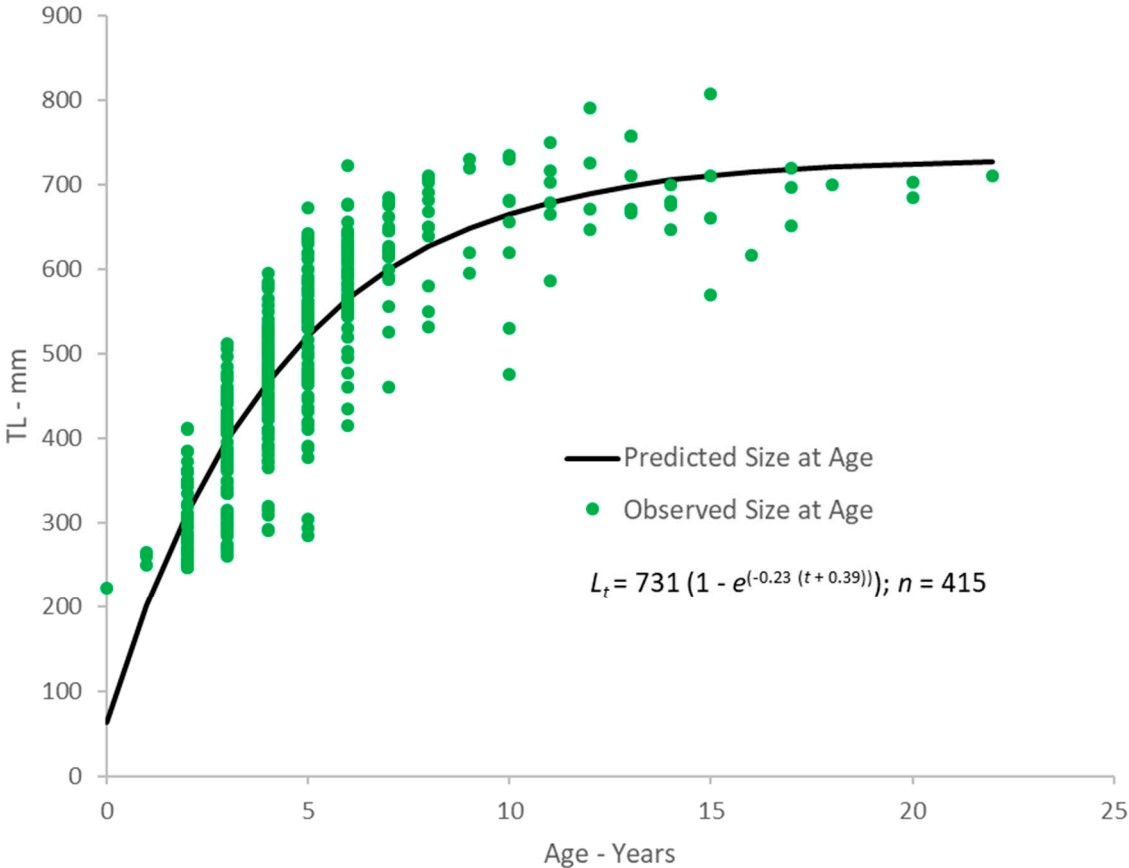

**Figure 4.** Comparison of observed size-at-calendar-age of margate (*Haemulon album*) to the predicted von Bertalanffy growth curve for fish from the Southeastern United States.

**Table 2.** Observed and predicted mean total length (TL) from the von Bertalanffy growth model, measured in millimeters, and natural mortality at calendar-age [18] data for black margate (*Anisotremus surinamensis*) collected from 1979 to 2017 from the east coast of Florida. Standard errors of the means (SE) are provided in parentheses.

| Age | *n* | Mean Observed TL (±SE) | TL Range | Predicted TL | $M\text{-}y^{-1}$ |
|-----|-----|------------------------|----------|--------------|--------|
| 0 | – | – | – | 0 | 7.20 |
| 1 | – | – | – | 84 | 1.57 |
| 2 | – | – | – | 155 | 0.83 |
| 3 | 1 | 342 | – | 215 | 0.56 |
| 4 | 6 | 295 (19) | 241–367 | 264 | 0.43 |
| 5 | 11 | 336 (14) | 291–451 | 306 | 0.36 |
| 6 | 10 | 353 (11) | 295–397 | 341 | 0.31 |
| 7 | 24 | 391 (16) | 311–628 | 370 | 0.28 |
| 8 | 20 | 409 (17) | 309–562 | 395 | 0.26 |
| 9 | 18 | 408 (19) | 292–595 | 416 | 0.24 |
| 10 | 15 | 411 (19) | 320–570 | 433 | 0.23 |
| 11 | 7 | 502 (29) | 377–592 | 448 | 0.22 |
| 12 | 3 | 429 (65) | 343–556 | 460 | 0.21 |
| 13 | 6 | 494 (50) | 355–615 | 470 | 0.20 |
| 14 | 1 | 354 | – | 478 | 0.20 |
| 15 | 2 | 576 (66) | 510–641 | 485 | 0.19 |
| 16 | 2 | 462 (41) | 421–502 | 492 | 0.19 |
| 17 | 2 | 485 (65) | 420–550 | 497 | 0.19 |
| 18 | 2 | 479 (34) | 445–512 | 501 | 0.19 |

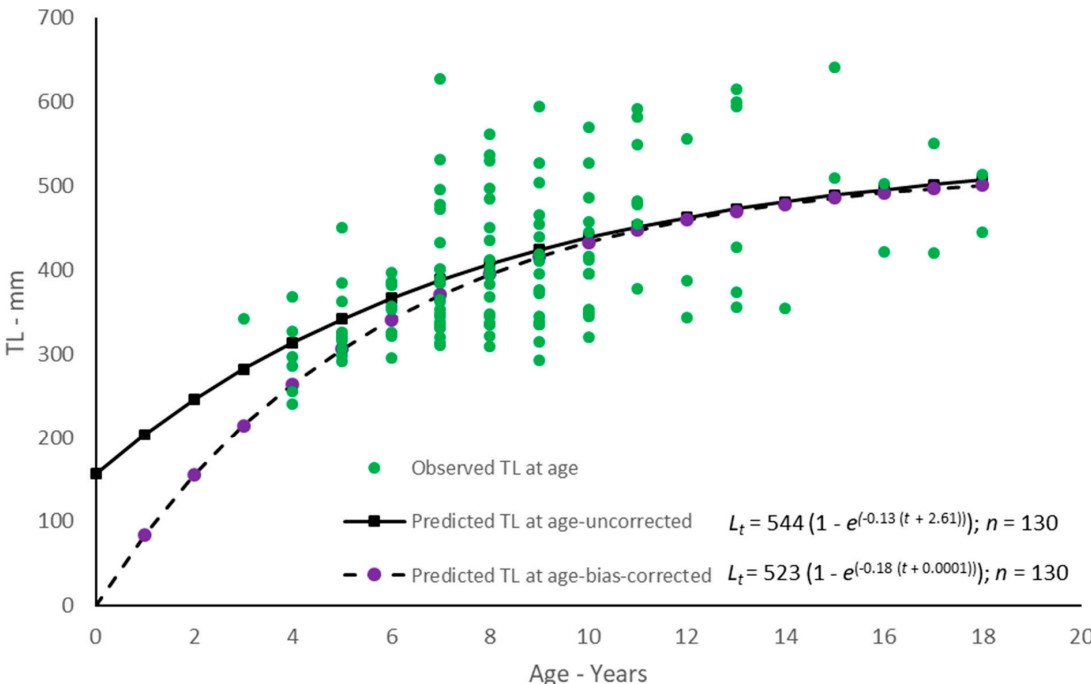

**Figure 5.** Comparison of observed size-at-calendar-age of black margate (*Haemulon album*) to the predicted von Bertalanffy growth curves for fish from the Southeastern United States. The bias-corrected model was run due to the lack of younger fish in our samples.

Because our data included no fish aged 0–2 years, and only a single age-3 fish, we re-estimated growth using a bias-correction procedure [19], which simulated smaller fish length at age for the youngest ages. The resulting parameter estimates were $L_\infty$ = 523 (32, 455–592), $k$ = 0.18 (0.03, 0.12–0.23), and $t_0$ = 0.0001 (0.004, −0.0071–0.0073); $n$ = 130, Figure 5. Black margate growth was slower than that of margate, attaining a mean observed length of 295 mm TL by age-4. Growth slowed substantially by age-13, after which the average yearly increase in size was 5 mm.

*2.3. Body-Size Relationships*

Statistical analyses revealed a multiplicative error term (variance increasing with size) in the residuals of the *W*-TL and *W*-FL relationship for both species, indicating a linearized ln-transform fit of the data was appropriate. The weight–length relationships for margate are described by the following regressions:

$$\ln(W) = -10.44 + 2.88 \ln(TL) \ (n = 1327, r^2 = 0.97; p < 0.0001; SE(a) = 0.08; SE(b) = 0.02)$$

and

$$\ln(W) = -10.27 + 2.90 \ln(FL) \ (n = 478, r^2 = 0.99; p < 0.0001; SE(a) = 0.09; SE(b) = 0.02)$$

We adjusted the intercept for log-transformation bias with the addition of one-half of the mean square error (1/2 MSE) [20] and transformed the equations back to the form $W = a(L)^b$, resulting in the equations (Figure 6).

$$W = 2.94 \times 10^{-5} \ TL^{2.88} \ (MSE = 0.02), \text{ and } W = 3.49 \times 10^{-5} \ FL^{2.90} \ (MSE = 0.02).$$

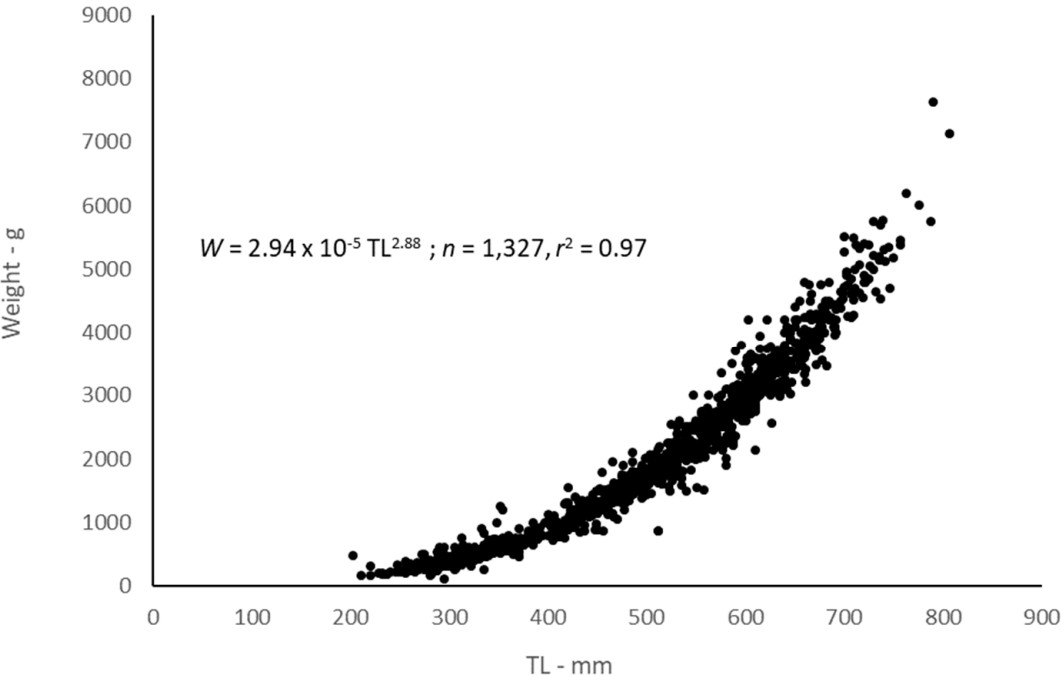

**Figure 6.** Weight–total-length relationship for margate, *Haemulon album*, for all fish measured by the Southeast Region Headboat Survey from 1979 to 2018.

Length–length relationships for margate are represented by the following equations:

$$\text{TL} = 1.1 \text{ FL} + 11.29 \text{ and FL} = 0.90 \text{ TL} - 7.6 \ (r^2 = 0.99, n = 513)$$

Weight–length relationships for black margate were:

$$\ln(W) = -11.10 + 3.02 \ln(\text{TL}) \ (n = 451, r^2 = 0.95; p < 0.0001; \text{SE(a)} = 0.19; \text{SE(b)} = 0.03)$$

and

$$\ln(W) = -10.21 + 2.93 \ln(\text{FL}) \ (n = 289, r^2 = 0.97; p < 0.0001; \text{SE(a)} = 0.18; \text{SE(b)} = 0.03)$$

Adjusting the intercept for log-transformation bias with the addition of one-half of the mean square error (1/2 MSE) [20] and transforming the equations back to the form $W = a(L)^b$ resulted in the following relationships (Figure 7):

$$W = 1.52 \times 10^{-5} \text{ TL}^{3.02} \ (\text{MSE} = 0.02) \text{ and } W = 3.69 \times 10^{-5} \text{ FL}^{2.93} \ (\text{MSE} = 0.01)$$

Length–length relationships for black margate were:

$$\text{TL} = 1.11 \text{ FL} + 7.05 \text{ and FL} = 0.89 \text{ TL} - 2.68 \ (r^2 = 0.99, n = 283)$$

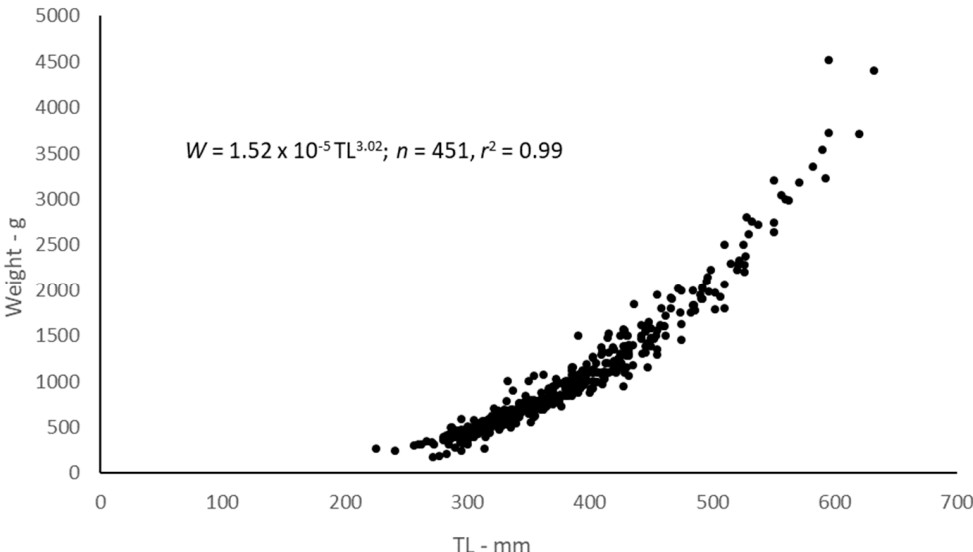

**Figure 7.** Weight–total-length relationship for black margate, *Anisotremus surimanensis*, for all fish measured by the Southeast Region Headboat Survey from 1972 to 2018.

### *2.4. Natural Mortality*

Using an age-invariant method integrating all ages into a single point estimate [21], natural mortality ($M$) was estimated to be 0.19 $y^{-1}$ and 0.23 $y^{-1}$ for margate and black margate, respectively, using their maximum ages of 22 years and 18 years in this study. Age-specific estimates of $M$ using a method that incorporates life history parameters [14] are presented in Tables 1 and 2. For these values, we used the midpoint of each age (e.g., 0.5, 1.5, 2.5, etc.), because the midpoint better represents the average annual size.

Cumulative survivorship to the oldest age for fully recruited ages, based on the age-varying $M$ [18], was 0.9% and 9.8% for margate (ages 5–22) and black margate (ages 8–18), respectively. When all ages are included, cumulative survivorship was 0.003% and 0.0001%, respectively, for the two species.

Estimating survivorship based on the point estimate of $M$ [21] will omit a large amount of mortality on the youngest ages and thus, will usually result in a greater proportion of the population surviving to the oldest age. Estimates of survivorship to the oldest age for margate were 3.3% for fully recruited ages and 1.3% for all ages. Estimates of black margate survivorship based on a static $M$ were 7.9% and 1.3% for the fully recruited ages and all ages, respectively.

### 3. Discussion

Otolith edge analysis demonstrated that margate and black margate each deposited one annulus per year from January through June, with the occurrence of otoliths with opaque margins being greatest in April and March, respectively. These results compare to findings of a previous study that found white grunt, *Haemulon plumieri*, deposited opaque zones from March to June, with occurrence of otoliths with opaque margins being greatest in May, for fish from Southeast Florida [22]. Similarly, a study from the Central Gulf Coast of Florida found white grunt also exhibited maximum occurrence of otoliths with opaque margins in May [23].

Margate grew rapidly, attaining an average observed size of 222 mm by age-1 and an average size of 520 mm by age-5. Margate from the SEUS appear to grow faster than margate from the southwest Cuban shelf, in which back-calculated lengths from otolith measurements were 181 mm and 481 mm for ages 1 and 5, respectively [14]. The only other published study on margate growth used scale measurements to estimate growth of margate from Jamaican fisheries [13].

Black margate grew somewhat slower than margate, attaining an average size of 336 mm by age-5. While margate attained 75% of $L_\infty$ by age-5, black margate did not achieve 75% of $L_\infty$ until age-7. This

difference in growth rate is substantiated by the values of $k$, the von Bertalanffy growth coefficient, for the two species: 0.23 $y^{-1}$ for the faster growing margate versus 0.18 $y^{-1}$ for black margate.

Estimated growth parameters for margate from our study agreed well with the previously published study using otoliths [10]: $L_\infty$ = 731 mm vs. 730 mm; $k$ = 0.18 $y^{-1}$ vs. 0.19 $y^{-1}$; $t_0$ = −0.38 y vs. −3.01 y. The predicted growth curve fit the observed data well (Figure 4), and the presence of the youngest ages in our samples convinced us that using a bias-correction procedure for estimating growth in margate was unnecessary.

There were no previously published studies of black margate life history in the literature for comparison with our study. Our initial predicted growth curve fit the observed data for the ages in our sample and could be used to estimate yield from fisheries landings, but our sample contained only one fish ≤age-3 to inform the early trajectory of the growth curve, resulting in a large value for $t_0$. This lack of younger fish was likely due to the fact that our study was comprised wholly of fishery-dependent samples, and many of the younger, smaller fish were not available in the landings. A few possible reasons include the following: (1) the small fish do not recruit to the gear used; (2) anglers do not keep the small fish when looking for larger sized fish or more desirable species due to regulatory trip limits; or (3) the small fish are not yet present in the habitat being fished. In addition, the number of samples available for this study was limited. The limits could be due to the low abundance of the species in the area or the fact that port agents (samplers working for federal and state natural resource agencies) tend to preferentially sample more economically important species to the reef fish fishery in the U.S. Atlantic waters. To estimate a more biologically reasonable growth curve for the population, we used a bias-correction method [19] to re-estimate growth, which adjusts for the bias imposed by minimum size limits by assuming zero probability of capture below the minimum size limit. Although there are no size limits for black margate, we set the functional size limit equal to the smallest fish in our sample, 241 mm. This procedure had the effect of pulling down the initial trajectory of the growth curve (Figure 5), and the resulting growth model yielded a more realistic size-at-age for the youngest fish.

We did not have enough samples to analyze whether there were inter-annual patterns in growth. Inter-annual variations in growth patterns might be caused by internal factors such as density dependence, or external factors such as temperature or fishing selectivities. While this type of analysis would be valuable, it is currently doubtful that, even with increased sampling intensity, sufficient landings of either of these less common species would be able to be procured in order to accomplish such an analysis.

Natural mortality of wild populations of fish is difficult to measure but is an important component of stock assessments. Many different methods to estimate $M$ based on life history parameters are available, but the most-used methods in SEDAR have been a static value based on maximum age and/or age-varying values based on the size of the fish using population growth model parameters. Although we offer an age-invariant estimate of $M$, we would expect $M$ to be much higher on the youngest, smallest fish compared to the fully recruited fish. The age-varying $M$ [18] seems a more appropriate estimator for the younger ages of both species. By the age at which margate are fully exploited in the fishery, age-5, the natural mortality rate at each age has dropped significantly and is approaching the point estimate we presented. For a fish that can attain the maximum size that margate can, we believe that the estimates of age-specific $M$ on the oldest ages are reasonable. In contrast, there is more uncertainty in the population growth parameters for black margate, due to the lack of fish younger than 3 years in our sample. However, the age-varying estimates of $M$ seem reasonable for this species. Because black margate is a smaller fish compared to margate, the initial values of $M$ for the youngest ages were higher, but then drop slightly lower than values for margate by age-6. Even with the limited samples for this study, the life history parameters we present are within reasonable range compared to other reef fish species with more extensive data.

This study provides the first reported analyses of margate and black margate life histories from SEUS waters, and the first reporting of black margate life history parameters from any geographic region. We have shown that otolith sections of both species contain annuli that can be used for ageing,

and timing of opaque zones deposition is similar for both species, with peak annulus formation in the spring months. Our estimates of *M* seem reasonable for species with moderate life spans. We believe the results of this study accurately describe growth of the fished populations of margate and black margate in the SEUS. While these two species are less commonly caught than many other species in the SEUS reef fishery, the examination and reporting of life history demographic parameters for previously unstudied species no doubt has value for ecosystem-based fishery management efforts.

## 4. Materials and Methods

### 4.1. Age Determination and Timing of Opaque Zone Formation

Margate (*n* = 415) and black margate (*n* = 130) samples were collected dockside by NMFS and state agencies sampling landings from the recreational and commercial sectors along the SEUS coast from 1979 to 2017. All specimens used in this study were killed as part of legal fishing operations and were already dead when sampled by the port agents; thus all research was conducted in accordance with the Animal Welfare Act (AWA) and with the US Government Principles for the Utilization and Care of Vertebrate Animals Used in Testing, Research, and Training (USGP) OSTP CFR, May 20 1985, Vol. 50, No. 97. All specimens were captured by either conventional vertical hook and line gear or divers with spears. Total lengths (TL) and/or fork lengths (FL) of specimens were recorded in millimeters (mm). Whole weight (*W*, grams) was recorded for fish landed in the recreational headboat fishery, while fish landed by commercial fisheries were eviscerated at sea and weights were not available.

Sagittal otoliths were removed during dockside sampling and stored dry. Otoliths were sectioned on a low-speed saw, resulting in three serial 0.5 mm sections, one on either side of, and one including, the otolith core. The sections were mounted on microscope slides with thermal cement and covered with mounting medium before analysis [24]. The sections were viewed under a dissecting microscope at 12.5× using transmitted light. Opaque zone counts were assigned to each sample by an experienced biologist [25,26]. Sections were read with no knowledge of date of capture or fish size. A subset of the otolith sections for each species was read by a second reader (margate: *n* = 194, or 47%; black margate: *n* = 27, or 21%). We calculated an index of average percent error (IAPE) [27]. When readers disagreed on a count, the sectio were viewed by both readers together. If a final age determination could not be made at that point, the sample was eliminated from the analysis.

Timing of opaque zone formation for each species was assessed using the relative distance between the outermost opaque zone and the otolith edge [16]. The edge types were plotted by month of capture to determine if the opaque zones were deposited primarily in one season or month. Based upon edge frequency analysis, all samples were assigned a calendar age, obtained by increasing the opaque zone count by one if the fish was caught before that year's increment was formed and had an edge which was a moderate to wide translucent zone. Fish caught during the time of year of opaque zone formation with a narrow or no translucent zone, as well as fish caught after the time of opaque zone formation, were assigned a calendar age equivalent to the opaque zone count.

### 4.2. Growth

We fitted the length-at-age (calendar age) data using the von Bertalanffy growth equation,

$$L_t = L_\infty\left(1 - e^{-k(t-t_0)}\right)$$

where $L_t$ is length at age $t$, $L_C$ is the theoretical maximum length (mm), $k$ is the Brody growth coefficient, or the rate at which maximum size is attained ($y^{-1}$), and $t_0$ is the theoretical age (in years, y) at length equal to 0 [17].

We anticipated there would be few fish of the youngest age classes available to us, as our samples were primarily fishery-dependent, and hook-and-line gear or fishers generally selected for older or larger fish. Consequently, the models would be unable to depict initial growth of the youngest fish, leading to difficulty in accurately estimating size at the youngest ages. We, therefore, re-ran the growth

models using a procedure that adjusted for the bias imposed by the selectivity of the fisheries, by assuming a zero probability of capture below the smallest fish in our samples [19]. This procedure had the effect of pulling the growth curve down to simulate smaller fish at age for the youngest ages (ages that we had no samples for).

### *4.3. Body-Size Relationships*

For weight–length relationships, we regressed *W* on TL (margate: *n* = 1327; black margate: *n* = 451) and *W* on FL (margate: *n* = 478; black margate: *n* = 289) using all fish sampled for lengths and whole weights by the SRHS from 1975 to 2015. Initially, the SRHS only measured TL of the fish during dockside sampling. Starting in the early 2000s, the SRHS began taking TL and FL on all fish sampled. We examined both a non-linear fit, $W = a\,\text{TL}^{\,b}$, using nonlinear least squares estimation [28] and a linearized fit of the log-transformed data, $\ln(W) = a + b\ln(\text{TL})$. Residuals were examined to determine which regression provided the best fit to the data. For length–length relationships, we regressed TL on FL and FL on TL (*n* = 283) using linear regression.

### *4.4. Natural Mortality*

We estimated the instantaneous rate of natural mortality (*M*) using two methods. We first estimated *M* using an age-invariant method using maximum age [21]:

$$M = 4.22/t_{max}$$

where $t_{max}$ is the maximum fish age encountered in the sample. The second method for estimating M was an age-varying method currently in use by SEDAR stock assessment process [18]:

$$M_t = (L_t/L_\infty)^{-1.5}\,k$$

where $M_t$ is natural mortality rate at age *a*, $L_\infty$ and *k* are the von Bertalanffy growth equation parameters and $L_t$ is fish length at age *t*. This method incorporates life-history information via the growth parameters and is based upon evidence suggesting that *M* is inversely related to body size [18].

**Author Contributions:** Conceptualization, M.L.B. and A.D.O.; Data curation, A.D.O.; Formal analysis, M.L.B. and J.C.P.; Investigation, J.C.P. and A.D.O.; Methodology, M.L.B., J.C.P. and A.D.O.; Project administration, M.L.B. and J.C.P.; Resources, J.C.P.; Software, J.C.P.; Supervision, M.L.B.; Validation, M.L.B.; Visualization M.L.B. and J.C.P.; writing—Original draft, M.L.B.; writing—Review & editing, M.L.B., Jennifer Potts and A.D.O.

**Funding:** This research received no external funding.

**Acknowledgments:** We gratefully acknowledge the many NMFS headboat and commercial port samplers over the years whose efforts made this study possible. K. Sigfried and T. Kellison provided constructive comments that greatly improved the manuscript. The scientific results and conclusions, as well as any views and opinions expressed herein, are those of the authors and do not necessarily reflect those of any government agency.

**Conflicts of Interest:** The authors declare no conflict of interest.

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
