# Peer review of "Preliminary Estimates of Age, Growth and Natural Mortality of Margate, Haemulon album, and Black Margate, Anisotremus surinamensis, from the Southeastern United States"

_fishes, doi:10.3390/fishes4030044_

Round 1
Reviewer 1 Report
This paper is well written and only have minor editorial requirements
Corrections required
Please calculate the r2 values for the VBGF models and include in the text on lines 157, 174 and 178. this is to be consistent with the presentations of the LW models. You can use the following equation
R2=1-(∑(y-y ̂)2)/(∑(y-y ̅)2))
on line 447 in the references section a return is required to separate references 12 and 13.
line 469-470. the journal name "Bulletin of marine Science" is written in full change to the abbreviated form.
Moreover on this point the use of full stops in the abbreviated journal names is inconsistently applied through out the reference section. Please revise these to be consistent and in line with the requirements of the journal fishes.
Comments to consider
The color contrasts in figs 2-3 are difficult to interpret when printed in grey scale. Could you modify these by using pattern fills or gray-scale colors (eg black, 70% grey, 30% grey, white) Would the authors consider using descriptive words instead of numbers for the legend eg opaque edge, narrow, intermediate, wide? As this makes interpretation of the figure easier
Figures 4 and 5 the contrast between the model line and markers is difficult to see in greyscale can you make the model line all black. Also bring the model line in front of the data. this can be done easily in excel (i assume these are made in excel) by using the model plotted on a secondary axis and then deleting the secondary axis from the chart. this results in the model being placed in front of the markers. I would also suggest not having markers on a line that comes from a model as it is a predictive model and not an observed point.
figures 6 and 7 why has the model not been included? I would make these scatter plots to the same format as Figures 4 and 5
lines 259-260. Directly comparing the growth coefficient values as a proxy for growth rate is not good practice as k and Linf are auto correlated. It is better to compare mean size at age with a t test if you want to say the species grows faster/slower at a particular point in time. .
Author Response
Reviewer # 1.
Please calculate r2 values for VBGF models and include in text……These models are non-linear models and r2 values are not appropriate as such. Since it seems the reviewer is asking for us to present some measure of error in the parameter estimates, I have reworked the text and included the parameter estimate standard errors and 95% confidence intervals for each parameter.
Line 447, reference section – Done.
Line 469-470 – Reference section journal abbreviations – Fixed.
General reference section comments – While I am not sure what full stops means, I have gone through the reference section and made sure my naming and abbreviations were consistent throughout (e.g., U.S. incorrect, changed to U. S., etc.
Figures 2- and 3. Since this is an online journal and will appear in color, I do not feel it necessary to modify the color scheme or use pattern fills so that someone viewing a black and white printed copy can discern differences, to this end I have made the borders of the different sections of the histograms bolder so that the differences can be better seen. Also, descriptive words for the legend are given in the figure caption, we do not feel it necessary to further modify the figure legend since the caption is right there.
Figures 6 and 7 – Why has the model not been included? - I assume the reviewer is talking about the r2 = value. I have included the r2 values from the ln-ln transformation run on each graph to provide a measure of goodness of fit of the data.
Lines 259-260 – t-test instead of comparing growth parameter values
We were only interested in contrasting our estimated growth parameters from our study with those derived by the only other published study to date. We are not interested in comparing mean size at age between the two studies, primarily because we do not have the raw data from the Cuban study and it would thus be impossible to do.

Reviewer 2 Report
The paper by Burton et al. provides the basic age and growth data for two Haemulids, which are either understudied or have not been studied before. The Haemulids are an important group of sub-tropical/ tropical species that constitute considerable biomass in some ecosystems around the world. However, there are very few comprehensive studies on the biology of these fishes. Unfortunately, this study is very limited in its scope, but could be expanded with addition work. The introduction focuses on the fisheries for these species, but could do with the addition of an overview of Haemulid age and growth. As a person from outside the USA, this paper does not really draw me in as it is solely focused on USA waters. What about the fisheries for fishes in this family in other regions of the world? In the same way the discussion provides no real interesting points, no comparisons with other species (except for a passing comments line 249-250 and 254-255.
Did the authors find if there was a difference in the growth of females and males, particularly for margate?
ABSTRACT
The Abstract should be used to provide the reader with an overview of the study, including its relevance, key results and any implications of the findings of the study. As it stands, the Abstract almost reads like a results section, particularly the middle section. I would suggest that the weight-length relationships are completely removed from the Abstract, they are not that interesting.
Lines 16-17, “Opaque zones were annular, forming January–June for both species, with peaks in April for margate and March for black margate.” This sentence is note written correctly. Something like “Opaque zones were annular, forming from January–June for both species, with peaks in occurrence of otoliths with opaque margins in April for margate and March for black margate.”
Lines 18-19 “Margate observed ages ranged 0–22 yrs, and the largest fish measured 807 mm TL (total length). Black margate observed ages ranged 3–17 years, and the largest fish was 641 mm TL.” This sentence is note written correctly. Something like “The observed ages for Margate were 0–22 yrs, and the largest fish measured 807 mm TL (total length). Black margate ranged in age from 3–17 years, and the largest fish was 641 mm TL.”
INTRODUCTION
Line 39. No need to spell out mm, everyone should know what this represents.
The introduction does not provide any background information on the biology of Haemulids. I would have thought that it would be necessary to provide some information so that the reader is aware of the longevity of other species and their growth characteristics. This needs some work.
RESULTS
Lines 88-89. Remove “no sections were excluded from analyses due to illegibility”. This is not necessary as you go on to say “Opaque zones on sectioned otoliths of both species were legible” (line 97).
Line 103. Remove “A final age was determined for all samples, and none were excluded from further analyses.”. This is not necessary as you have already said “Opaque zones on sectioned otoliths of both species were legible” (line 97).
Lines 117-118. This needs better explanation, particularly what the peak is referring to. “Margate otoliths exhibited opaque zones on the margin January through June, with a peak in February ” change to “Margate otoliths exhibited opaque zones on the margin from January through to June, with a peak in the occurrence of such otoliths in February ”
Line 118. Insert “a” between “to” and “narrow”.
Lines 119-120. Again, needs far better explanation. “followed by an increase in moderate translucent edges in September, and wide translucent edge types increasing from October to January, when opaque edges begin to form.” change to “followed by an increase in the occurrence of sections with moderate translucent edges in September, and the occurrence of sections wide translucent edge types increasing from October to January, when opaque edges begin to form.”
Lines 125-126. “The amount of wide translucent edge types decreased from January through March” change to “The occurrence of sections with wide translucent edge types decreased from January through to March”
Lines 131 and 141. Remove “indicating annulus formation”
Line 146. Needs rewording. Change “opaque zone formation timing” to “the timing of opaque zone formation”.
Line 154. Change “Margate ranged in size from 221 – 807 mm TL and ages 0 – 22 years” to “Margate ranged in size from 221 – 807 mm TL and in age from 0 – 22 years”.
Line 155. Should read “..fish were older than 17 years of age”.
Figure 4 and 5. Remove von Bertalanffy equation(s) from within the figure, they are already provided in the text.
Lines 174 and 178. Italicise n.
Figure 6 and 7. Remove equation(s) from within the figure, they are already provided in the text.
DISCUSSION
Line 246. “… with peak annulus formation occurring in April and March.” The opaque zone does not form in a single month, but over many months. This should be reworded to say “… with the occurrence of otoliths with opaque margins being greatest in April and March.”
Line 247. What is “respectively” referring to. Should be deleted.
Line 248. “…peaking in May”. See my comment above.
Line 250. “ …exhibited peak annulus deposition in May”. See my comment above.
Line 259. Need to have units for the k values of 0.23 and 0.18.
Line 262. Need to have units for the k and t0 values.
Lines 282-284. The authors state that they believe the bias-corrected growth curve provides a more realistic size-at-age for the youngest black margate. While the curve may look biologically more realistic, by passing through zero, the curve actually passes towards the bottom end of the group points for those 4 and 5 year old fish. So the curve is in fact underestimating the length at age of these fish. I believe that this needs to be rethought, is it real providing a better fit and/or, should a growth curve be fitted to these data in the first place.
MATERIALS AND METHODS
Line 355. Need units for von Bertalanffy parameters.
Author Response
Reviewer # 2 – We have accepted the vast majority of the reviewer’s comments concerning rewording and have made those changes. The few exceptions to these and the reasons why follow:
General comments about introduction – This paper is mainly focused on the characteristics of these fishes from US waters, because that is where the samples were collected, and that is the management purview of the agency I work for. We feel the introduction provides a good introduction to the species studied in this paper, in terms of geographic distribution, previous biological studies, and importance to the local (SEUS fishery). While it may be true there is much more known about a limited number of other Haeumulid species, such as white grunt, we did not set out to write a review paper.
Figs 4 and 5, remove von bertalanffy equations from figures. This is not necessary. We feel each graph should be able to be a stand alone graph, displaying as much information as possible about the data presented within that graph. This is more important now with these revisions to display these equations in the graph since we have removed them from the body of the results in order to present the standard errors and confidence intervals, measures of uncertainty that the reviewer asked for.
Figures 6 and 7, remove equations from within figures. Again, no we don’t feel this is necessary, they add value to the figure and it doesn’t cost any extra!
Line 282-284. The authors state they believe the bias corrected growth curve provides a more realistic size at age for the youngest black margate. While the curve may look more biologically realistic, by passing through zero, the curve actually passes towards the bottom end of the group of points for those 4 and 5 year old fish. So the curve is in fact underestimating the length at age of these fish. I believe that this needs to be rethought, is it really (sp) providing a better fit and/or, should a growth curve be fitted to these data in the first place?
The premise for running the bias corrected growth curve is the assumption the overall size range of the fish at the youngest ages in our sample, e.g., ages 3-7 years for black margate, were truncated on the left side, or lower end, of the range (McGarvey and Fowler, 2002). In other words, we expected the growth curve to pass through the lower end of the size range of the 4 and 5 year olds, because the smaller fish at those ages were not retained in the catch. By the time the fish reached age-8 the growth models, bias corrected and no correction, begin to converge. This result suggests the size range of the fish at ages 8 – max age are no longer affected by the size selectivity bias. All of the parameters were freely estimated in the model, and we feel that the resulting growth model using the bias correction actually reflects the population better. We do provide the growth model for those fish retained in the fishery, which can be used to calculate yield in a stock assessment.